# Evidence for prevalent $Z = 6$ magic number in neutron-rich carbon isotopes

D.T. Tran[1,2], H.J. Ong [1], G. Hagen[3,4], T.D. Morris[3,4], N. Aoi[1], T. Suzuki[5,6], Y. Kanada-En'yo[7], L.S. Geng[8], S. Terashima[8], I. Tanihata[1,8], T.T. Nguyen[9,10,27], Y. Ayyad[1], P.Y. Chan[1], M. Fukuda[11], H. Geissel[12,13], M.N. Harakeh[12,14], T. Hashimoto[15], T.H. Hoang[1,2], E. Ideguchi[1], A. Inoue[1], G.R. Jansen [3,16], R. Kanungo[17], T. Kawabata[7], L.H. Khiem[2], W.P. Lin[18], K. Matsuta[11], M. Mihara[11], S. Momota[19], D. Nagae[20], N.D. Nguyen[21], D. Nishimura[22], T. Otsuka[23], A. Ozawa[24], P.P. Ren[18], H. Sakaguchi[1], C. Scheidenberger[12,13], J. Tanaka[1], M. Takechi[25], R. Wada[18,26] & T. Yamamoto[1]

The nuclear shell structure, which originates in the nearly independent motion of nucleons in an average potential, provides an important guide for our understanding of nuclear structure and the underlying nuclear forces. Its most remarkable fingerprint is the existence of the so-called magic numbers of protons and neutrons associated with extra stability. Although the introduction of a phenomenological spin–orbit (SO) coupling force in 1949 helped in explaining the magic numbers, its origins are still open questions. Here, we present experimental evidence for the smallest SO-originated magic number (subshell closure) at the proton number six in $^{13-20}$C obtained from systematic analysis of point-proton distribution radii, electromagnetic transition rates and atomic masses of light nuclei. Performing ab initio calculations on $^{14,15}$C, we show that the observed proton distribution radii and subshell closure can be explained by the state-of-the-art nuclear theory with chiral nucleon–nucleon and three-nucleon forces, which are rooted in the quantum chromodynamics.

[1] Research Center for Nuclear Physics, Osaka University, Osaka 567-0047, Japan. [2] Institute of Physics, Vietnam Academy of Science and Technology, Hanoi 10000, Vietnam. [3] Physics Division, Oak Ridge National Laboratory, Oak Ridge, TN 37831, USA. [4] Department of Physics and Astronomy, University of Tennessee, Knoxville, TN 37996, USA. [5] Department of Physics, College of Humanities and Sciences, Nihon University, Tokyo 156-8550, Japan. [6] National Astronomical Observatory of Japan, Tokyo 181-8588, Japan. [7] Department of Physics, Kyoto University, Kyoto 606-8502, Japan. [8] School of Physics and Nuclear Energy Engineering, Beihang University, 100191 Beijing, China. [9] Pham Ngoc Thach University of Medicine, Ho Chi Minh 700000, Vietnam. [10] Faculty of Physics and Engineering, VNUHCM-University of Science, Ho Chi Minh City 70250, Vietnam. [11] Department of Physics, Osaka University, Osaka 560-0043, Japan. [12] GSI Helmholtzzentrum für Schwerionenforschung, 64291 Darmstadt, Germany. [13] Justus Liebig University, 35392 Giessen, Germany. [14] KVI Center for Advanced Radiation Technology, University of Groningen, 9747 AA Groningen, The Netherlands. [15] Rare Isotope Science Project, Institute for Basic Science, Daejeon 34047, Korea. [16] National Center for Computational Sciences, Oak Ridge National Laboratory, Oak Ridge, TN 37831, USA. [17] Astronomy and Physics Department, Saint Mary's University, Halifax, NS B3H 3C3, Canada. [18] Institute of Modern Physics, Chinese Academy of Sciences, 730000 Lanzhou, China. [19] Kochi University of Technology, Kochi 782-8502, Japan. [20] RIKEN Nishina Center, Saitama 351-0198, Japan. [21] Dong Nai University, Dong Nai 81000, Vietnam. [22] Tokyo University of Science, Chiba 278-8510, Japan. [23] Department of Physics, University of Tokyo, Tokyo 113-0033, Japan. [24] Institute of Physics, University of Tsukuba, Ibaraki 305-8571, Japan. [25] Department of Physics, Niigata University, Niigata 950-2181, Japan. [26] Cyclotron Institute, Texas A&M University, College Station, TX 77840, USA. [27] Present address: Sungkyunkwan University, Gyeonggi-do 16419, South Korea. Correspondence and requests for materials should be addressed to H.J.O. (email: onghjin@rcnp.osaka-u.ac.jp)

Atomic nuclei—the finite quantum many-body systems consisting of protons and neutrons (known collectively as nucleons)—exhibit shell structure, in analogy to the electronic shell structure of atoms. Atoms with filled electron shells—known as the noble gases—are particularly stable chemically. The filling of the nuclear shells, on the other hand, leads to the magic-number nuclei. The nuclear magic numbers, as we know in stable and naturally-occurring nuclei, consist of two different series of numbers. The first series—2, 8, 20—is attributed to the harmonic-oscillator (HO) potential, while the second one—28, 50, 82 and 126—is due to the spin–orbit (SO) coupling force (see Fig. 1). It was the introduction of this phenomenological SO force—a force that depends on the intrinsic spin of a nucleon and its orbital angular momentum, and the so-called $j$–$j$ coupling scheme that helped explain[1,2] completely the magic numbers, and won Goeppert-Mayer and Jensen the Nobel Prize. However, the microscopic origins of the SO splitting have remained unresolved due to the difficulty to describe the structure of atomic nuclei from ab initio nuclear theories[3–5] with two- (NN) and three-nucleon forces (3NFs). Although the theoretical study[6] of the SO splitting of the $1p_{1/2}$ and $1p_{3/2}$ single-particle states in $^{15}$N has suggested possible roles of two-body SO and tensor forces, as well as three-body forces, the discovery of a prevalent SO-type magic number 6 is expected to offer unprecedented opportunities to understand its origins.

In her Nobel lecture, Goeppert-Mayer had mentioned the magic numbers 6 and 14—which she described as hardly noticeable—but surmised that the energy gap between the $1p_{1/2}$ and $1p_{3/2}$ orbitals due to the SO force is fairly small[7]. That the $j$–$j$ coupling scheme appears to fail in the $p$-shell light nuclei was discussed and attributed to their tendency to form clusters of nucleons[8]. Experimental and theoretical studies in recent decades, however, have hinted at the possible existence of the magic number 6 in some semimagic unstable nuclei, each of which has a HO-type magic number of the opposite type of nucleons. For instance, possible subshell closures have been suggested in $^{8}$He[9–11], $^{14}$O[12] and $^{14}$C[12–14]. Whether the subshell closure at the proton number $Z = 6$ is predominantly driven by the shell closure at the neutron number $N = 8$ in $^{14}$C or persists in other carbon isotopes is of fundamental importance.

The isotopic chain of carbon—with six protons and consisting of thirteen particle-bound nuclei—provides an important platform to study the SO splitting of the $1p_{1/2}$ and $1p_{3/2}$ orbitals. Like other lighter isotopes, the isotopes of carbon are known to exhibit both clustering[15–17] and single-particle behaviours. Although the second excited $J^\pi = 0^+$ state in $^{12}$C—the famous Hoyle state and important doorway state that helps produce $^{12}$C in stars—is well understood as a triple-alpha state, it seems that the effect of the alpha-cluster-breaking $1p_{3/2}$ subshell closure is important to reproduce the ground-state binding energy[18]. For even–even neutron-rich carbon isotopes, theoretical calculations using the anti-symmetrized molecular dynamics (AMD)[19], shell model[20,21], as well as the ab initio no-core shell model calculation[22] with NN + 3NFs have predicted near-constant proton distributions, a widening gap between proton $1p_{1/2}$ and $1p_{3/2}$ single-particle orbits, and a remarkably low proton occupancy in the $1p_{1/2}$ orbit, respectively. Gupta et al.[23], on the other hand, have suggested the possible existence of closed-shell core nuclei in $^{15,17,19}$C on the basis of potential energy surfaces employing the cluster-core model. Experimentally, small $B(E2)$ values comparable to that of $^{16}$O were reported from the lifetime measurements of the first excited $2^+$ $\left(2_1^+\right)$ states in $^{16,18}$C[24–26]. The small $B(E2)$ values indicate small proton contributions to the transitions, and together with the theoretical predictions may imply the existence of a proton-subshell closure.

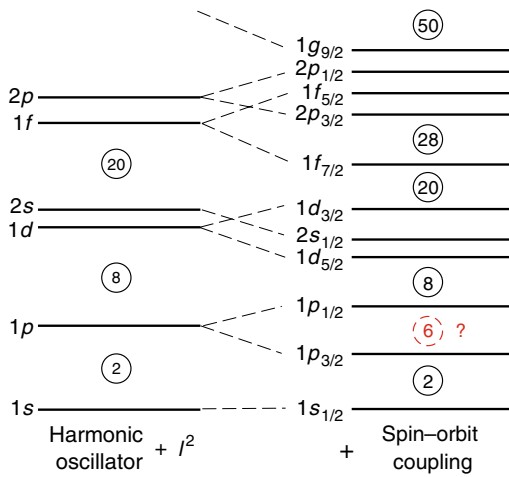

**Fig. 1** Nuclear shell structure. The left diagram is the shell structure for a harmonic-oscillator potential plus a small orbital angular momentum ($l^2$) term. The right diagram shows the splitting of the single-particle orbitals by an additional spin–orbit coupling force

Although still not well established, the size of a nucleus, which can be defined as the root-mean-square (rms) radius of its nucleon distribution, is expected to provide important insights on the evolution of the magic numbers. Recently, an unexpectedly large proton rms radius (denoted simply as proton radius hereafter) was reported[27], and suggested as a possible counter-evidence for the double shell closure in $^{52}$Ca[28]. Attempts to identify any emergence of non-traditional magic numbers based on the analysis of the systematics of the experimental proton radii have been reported[12,29]. For the $4 < Z < 10$ region, the lack of experimental data on the proton radii of neutron-rich nuclei due to the experimental and theoretical limitations of the isotope-shift method has hindered systematic analysis of the radii behaviour. Such systematic analysis has become possible very recently owing to the development of an alternative method to extract the proton radii of neutron-rich nuclei from the charge-changing cross-section measurements.

Here we present experimental evidence for a prevalent $Z = 6$ subshell closure in $^{13-20}$C, based on a systematic study of proton radii obtained from our recent experiments as well as the existing nuclear charge radii[12], electric quadrupole transition rates $B(E2)$ between the $2_1^+$ and ground $\left(0_{gs}^+\right)$ states of even–even nuclei[30], and atomic-mass data[31]. We show, by performing coupled-cluster calculations, that the observations are supported by the ab initio nuclear model that employs the nuclear forces derived from the effective field theory of the quantum chromodynamics.

## Results

**Experimental details.** The charge-changing cross section (denoted as $\sigma_{CC}$) of a projectile nucleus on a nuclear/proton target is defined as the total cross section of all processes that change the proton number of the projectile nucleus. Applying this method, we have determined the proton radii of $^{14}$Be[32], $^{12-17}$B[33] and $^{12-19}$C[34,35] from the $\sigma_{CC}$ measurements at GSI, Darmstadt, using secondary beams at around 900 MeV per nucleon. In addition, we have also measured $\sigma_{CC}$'s for $^{12-18}$C on a $^{12}$C target with secondary beams at around 45 MeV per nucleon at the exotic nuclei (EN) beam line[36] at RCNP, Osaka University. To extract proton radii from both low-energy data and high-energy data, we have devised a global parameter set for use in the Glauber-model calculations. The Glauber model thus formulated was applied to

**Table 1 Cross sections and proton radii**

| | $E_{CC}$ (A MeV) | $\sigma_{CC}$ (mb) | $E_{CC}$ (A MeV) | $\sigma_{CC}$ (mb) | $R_p$ (fm) |
|---|---|---|---|---|---|
| $^{17}$C | 46.3 | 1000(16) | 979 | 754(7) | 2.43(4) |
| $^{18}$C | 42.8 | 1023(31) | 895 | 747(7) | 2.42(5) |
| $^{19}$C | | | 895 | 749(9) | 2.43(4) |

Measured charge-changing cross sections ($\sigma_{CC}$) for $^{17-19}$C and the corresponding secondary-beam energies ($E_{CC}$). The subscript CC denotes the charge-changing reaction. The data in the fourth and fifth columns are from ref. [34] $R_p$'s in the sixth column are the proton radii extracted from the $\sigma_{CC}$'s in the third and fifth columns

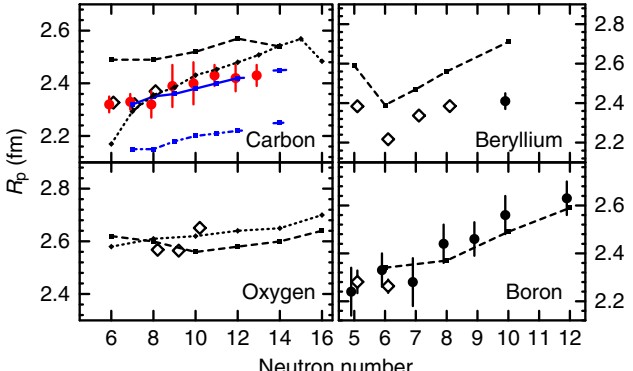

**Fig. 2** Proton radii. Results are shown for carbon, beryllium, boron and oxygen isotopes. The red-filled and black-filled circles are, respectively, the proton radii from this and our recent work[32–34,37]. The open diamonds are the data from electron-scattering and isotope-shift methods[12]. The error bars for the red-filled circles include the statistical and experimental systematic uncertainties, as well as the uncertainties due to the choice of density distributions. The error bars for other experimental data are taken from the literature. The small symbols connected with dashed and dotted lines are the predictions from the AMD[19] and RMF[38] models, respectively. The small blue symbols with solid and dash-dotted lines are the results from the ab initio coupled-cluster calculations with NNLO$_{sat}$[39] and the NN-only interaction NNLO$_{opt}$[40]

the $\sigma_{CC}$ data at both energies to determine the proton radii. A summary on the experiment at RCNP and the Glauber-model analysis is given in Methods. More details can be found in ref. [37]

**Charge-changing cross sections and proton radii.** For simplicity, we show only the results for $^{17-19}$C in Table 1; for results on $^{12-16}$C, see ref. [37] $R_p$'s are the proton radii extracted using the Glauber model formulated in ref. [37] The values for $^{17,18}$C are the weighted mean extracted using $\sigma_{CC}$'s at the two energies, while the one for $^{19}$C was extracted using the high-energy data. In determining the proton radii, we have assumed harmonic-oscillator (HO)-type distributions for the protons in the Glauber calculations. The uncertainties shown in the brackets include the statistical uncertainties, the experimental systematic uncertainties, and the uncertainties attributed to the choice of functional shapes, that is HO or Woods–Saxon, assumed in the calculations.

To get an overview of the isotopic dependence, we compare the proton radii of the carbon isotopes with those of the neighbouring beryllium, boron and oxygen isotopes. Figure 2 shows the proton radii for carbon, beryllium, boron and oxygen isotopes. The red-filled and black-filled circles are the data for $^{12-19}$C, beryllium and boron isotopes extracted in this and our previous work[32–34,37]. For comparison, the proton radii determined with the electron-

scattering and isotope-shift methods[12] are also shown in Fig. 2 (open diamonds). Our $R_p$'s for $^{12-14}$C are in good agreement with the electron-scattering data. In addition, we performed theoretical calculations. The small symbols connected with dashed and dotted lines shown in the figure are the results from the AMD[19] and relativistic mean field (RMF)[38] calculations, respectively. The blue-solid and blue-dash-dotted lines are the results (taken from ref. [34]) of the ab initio coupled-cluster (CC) calculations with NNLO$_{sat}$[39] and the NN-only interaction NNLO$_{opt}$[40], respectively. The AMD calculations reproduce the trends of all isotopes qualitatively but overestimate the proton radii for carbon and beryllium isotopes. The RMF calculations, on the other hand, reproduce most of the proton radii of carbon and oxygen isotopes but underestimate the one of $^{12}$C. Overall, the CC calculations with the NNLO$_{sat}$ interactions reproduce the proton radii for $^{13-18}$C very well. The calculations without 3NFs underestimate the radii by about 10%, thus suggesting the importance of 3NFs.

It is interesting to note that $R_p$'s are almost constant throughout the isotopic chain from $^{12}$C to $^{19}$C, fluctuating by less than 5%. Whereas this trend is similar to the one observed/predicted in the proton-closed shell oxygen isotopes, it is in contrast to those in the beryllium and boron isotopic chains, where the proton radii change by as much as 10% (for berylliums) or more (for borons). It is also worth noting that most theoretical calculations shown predict almost constant proton radii in carbon and oxygen isotopes. The large fluctuations observed in Be and B isotopes can be attributed to the development of cluster structures, whereas the almost constant $R_p$'s for $^{12-19}$C observed in the present work may indicate an inert proton core, that is $1p_{3/2}$ proton-subshell closure.

**Systematics of nuclear observables.** Examining the $Z$ dependence of the proton rms radii along the $N = 8$ isotonic chain, Angeli et al. have pointed out[12,29] a characteristic change of slope (existence of a kink), a feature closely associated with shell closure, at $Z = 6$. Here, by combining our data with the recent data[32–34,37], as well as the data from ref. [12], we plot the experimental $R_p$'s against proton number. To eliminate the smooth mass number dependence of the proton rms radii, we normalised all $R_p$'s by the following mass-dependent rms radii[41]:

$$R_p^{cal} = \sqrt{3/5}\left(1.15 + 1.80A^{-2/3} - 1.20A^{-4/3}\right)A^{1/3}\,\text{fm}.$$

Figure 3a shows the evolution of $R_p/R_p^{cal}$ with proton number up to $Z = 22$ and for isotonic chains up to $N = 28$. Each isotonic chain is connected by a solid line. For simplicity, only the symbols for $N = 3–16$ are shown in the legend in Fig. 3c; the data for $N = 6–13$ isotones are displayed in colours for clarity. For nuclides with more than one experimental value, we have adopted the weighted mean values. The discontinuities observed at $Z = 10$ and $Z = 18$ are due to the lack of experimental data in the proton-rich region. Note the increase/change in the slope at the traditional magic numbers $Z = 8$ and 20. Marked kinks, similar to those observed at $Z = 20, 28, 50$ and $82$[29], are observed at $Z = 6$ for isotonic chains from $N = 7$ to $N = 13$, indicating a possible major structural change, for example emergence of a subshell closure, at $Z = 6$.

The possible emergence of a proton-subshell closure at $Z = 6$ in neutron-rich even–even carbon isotopes is also supported by the small $B(E2)$ values observed in $^{14-20}$C[25,26,30]. Figure 3b shows the systematics of $B(E2)$ values in Weisskopf unit (W.u.) for even–even nuclei up to $Z = 22$. All data are available in ref. [30]. Nuclei with shell closures manifest themselves as minima. Besides the traditional magic number $Z = 8$, clear minima with $B(E2)$

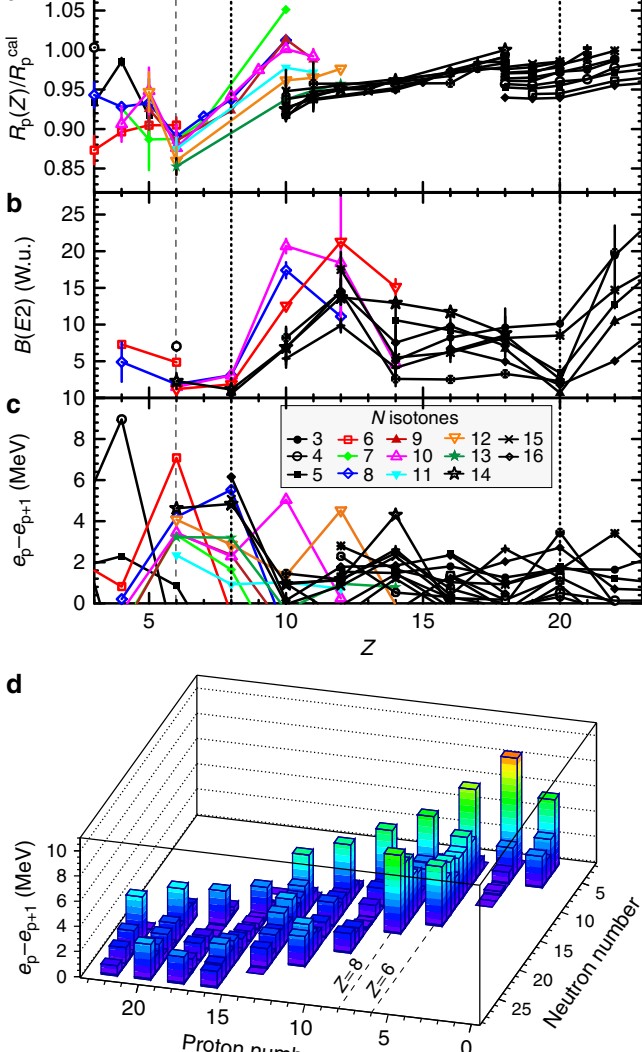

**Fig. 3** Systematics of nuclear observables. Evolution of **a** $R_p/R_p^{cal}$, **b** $B(E2)$ and **c** $e_p - e_{p+1}$ with proton number up to $Z = 22$ and for isotonic chains up to $N = 28$. Vertical dotted and thin-dashed lines denote positions of the traditional proton magic numbers and $Z = 6$, respectively. The error bars for data in **a** are evaluated using the errors, while the ones in **b** are the errors from the literature. For clarity, the error bars in **c**, some of which are slightly larger than the symbols, are not shown. **d** Two-dimensional lego plot of **c**

values smaller than 3 W.u. are observed at $Z = 6$ for $N = 8$, 10, 12 and 14 isotones.

To further examine the possible subshell closure at $Z = 6$, we consider the second derivative of binding energies defined as follows[42]:

$$\Delta_p^{(3)}(N, Z) \equiv (-1)^Z [S_p(N, Z) - S_p(N, Z + 1)]/2, \quad (1)$$

where $S_p(N, Z)$ is the one-proton separation energy. In the absence of many-body correlations such as pairings, $S_p(N, Z)$ resembles the single-particle energy, and $2\Delta_p^{(3)}(N, Z)$ yields the proton single-particle energy-level spacing or shell gap between the last occupied ($e_p$) and first unoccupied proton orbitals ($e_{p+1}$) in the nucleus with $Z$ protons (and $N$ neutrons). To eliminate the effect of proton–proton ($p$–$p$) pairing, we subtract out the $p$–$p$ pairing energies using the empirical formula: $\Delta_p = 12A^{-1/2}$ MeV. Figure 3c shows the systematics of $e_p - e_{p+1}$ ($= 2\Delta_p^{(3)}(N, Z) - 2\Delta_p$) for even-$Z$ nuclides. All data were evaluated from the experimental

binding energies[31]. Here, we have omitted odd-$Z$ nuclides to avoid odd–even staggering effects. The cusps observed at $Z = N$ for all isotonic chains are due to the Wigner effect[43]. Apart from the $Z = N$ nuclides, sizable gaps (>2 MeV) are also observed at $Z = 6$ for $N = 7$–14, and at $Z = 8$ for $N = 8$–10 and 12–16. For clarity, we show the corresponding two-dimensional lego plot in Fig. 3d.

By requiring a magic nucleus to fulfil at least two signatures in Fig. 3a–c, we conclude that we have observed a prominent proton-subshell closure at $Z = 6$ in $^{13-20}$C. Although the empirical $2\Delta_p^{(3)}$ for $^{12}$C is large (~14 MeV), applying the prescription from ref. [44], we obtain about 10.7 MeV for the total $p$–$p$ and $p$–$n$ pairing energy. This estimated large pairing energy indicates possible significant many-body correlations such as cluster correlations. We note that $^{12}$C is known to be an intermediate-coupling nucleus lying in the middle of the $j$–$j$ coupling and $L$–$S$ coupling limits[45]. The core is largely broken with only about 40% of the nominal $(1p_{3/2})^8$ closed-shell component, and the occupation number of nucleons in the $1p_{1/2}$ shell is as much as 1.5 from shell model calculations using the Cohen–Kurath interactions[46].

## Discussion

It is surprising that the systematics of the proton radii, $B(E2)$ values and the empirical proton-subshell gaps for most of the carbon isotopes are comparable to those for proton-closed shell oxygen isotopes. To understand the observed ground-state properties, that are the proton radii and subshell gap of the carbon isotopes, we performed ab initio CC calculations on $^{14,15}$C using various state-of-the-art chiral interactions. We employed the CC method in the singles-and-doubles approximation with perturbative triples corrections [Λ-CCSD(T)][47] to compute the ground-state binding energies and proton radii for the closed-(sub)shell $^{14}$C. To compute $^{15}$C ($1/2^+$), we used the particle-attached equation-of-motion CC (EOM-CC) method[48], and included up to three-particle-two-hole (3p–2h) and two-particle-three-hole (2p–3h) corrections as recently developed in ref. [49] Figure 4 shows the binding energies as functions of the proton radii for (a) $^{14}$C and (b) $^{15}$C. The coloured bands are the experimental values; the binding energies (red-horizontal lines) are taken from ref. [31], while proton radii are from ref. [37] (orange bands) and the electron-scattering data[12] (green band). The filled black symbols are CC predictions with the NN + 3NF chiral interactions from ref. [50] labelled 2.0/2.0 (EM)(black square), 2.0/2.0 (PWA)(black downward-pointing triangle), 1.8/2.0 (EM) (black circle), 2.2/2.0 (EM)(black diamond), 2.8/2.0 (EM) (black triangle), and NNLO$_{sat}$[39] (black star). Here, the NN interactions are the next-to-next-to-next-to-leading order (N$^3$LO) chiral interaction from ref. [51], evolved to lower cutoffs (1.8/2.0/2.2/2.8 fm$^{-1}$) via the similarity-renormalisation-group (SRG) method[52], while the 3NF is taken at NNLO with a cutoff of 2.0 fm$^{-1}$ and adjusted to the triton binding energy and $^4$He charge radius. The error bars are the estimated theoretical uncertainties due to truncations of the employed method and model space. For details on the CC method and error estimation, see refs. [5,53]. Note that the error bars for the binding energies are smaller than the symbols. Depending on the NN cutoff, the calculated binding energy correlates strongly with the calculated proton radius. In addition, we performed the CC calculations with chiral effective interactions without 3NFs, that are the NN-only EM interactions with NN cutoffs at 1.8 (white circle), 2.0 (white square), 2.2 (white diamond) and 2.8 fm$^{-1}$ (white triangle), and the NN-only part of the chiral interaction NNLO$_{sat}$ (white downward-pointing triangle). Overall, most calculations that include 3NFs reproduce the experimental proton radii well. For the binding energies, the

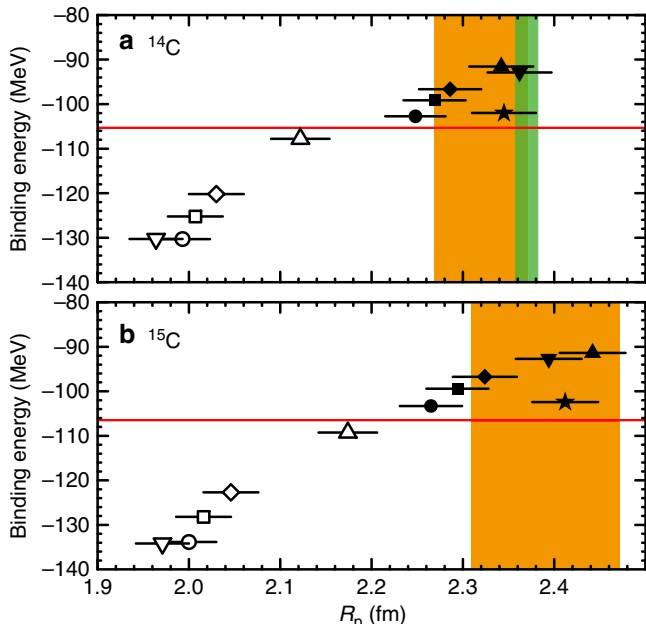

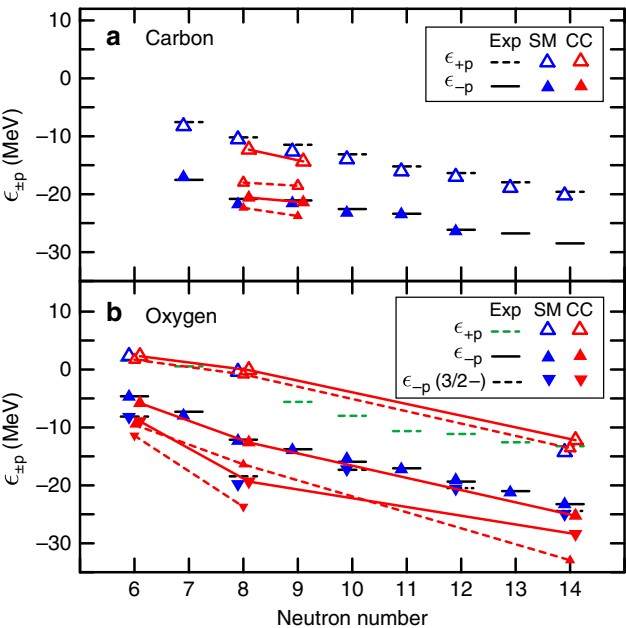

**Fig. 4** Binding energies versus proton radii. The results for **a** $^{14}$C and **b** $^{15}$C are shown. The coloured bands and red-horizontal lines are the experimental values. The green band represents the proton radius from the electron scattering. The filled black symbols are the CC predictions with SRG-evolved NN + 3NF chiral effective interactions at different NN/3NF cutoffs and NNLO$_{sat}$, whereas the open symbols are the predictions with the NN-only EM and NNLO$_{sat}$ interactions. The error bars are the estimated theoretical uncertainties due to truncations of the employed method and model space[53]. See text for details

**Fig. 5** Shell evolution. Empirical one-proton addition ($\epsilon_{+p}$) and removal ($\epsilon_{-p}$) energies (horizontal bars) for **a** carbon, and **b** oxygen isotopes deduced from one-proton separation energies and the excitation energies of the lowest $3/2^-$ states in the odd–even nitrogen isotopes. The dotted bars indicate the adopted values for the observed excited states in $^{19,21}$N, which have been tentatively assigned as $3/2^{-57}$. Other experimental data are taken from refs. [31,58,59]. The blue symbols are the shell model calculations using the YSOX interactions[60]. Results of the CC calculations with and without 3NFs are shown by the red-solid and red-dashed lines, respectively

calculations with the EM(1.8/2.0) and NNLO$_{sat}$ interactions reproduce both data very well. It is important to note that without 3NFs the calculated proton radii are about 9–15% (18%) smaller, while the ground states are overbound by as much as about 24% (26%) for $^{14}$C ($^{15}$C). These results highlight the importance of comparing both experimental observables to examine the employed interactions.

The importance of the Fujita–Miyazawa type[54] or the chiral NNLO 3NFs[55,56] in reproducing the binding energies and the drip lines of nitrogen and oxygen isotopes have been suggested in recent theoretical studies. Here, to shed light on the role of 3NFs on the observed subshell gap, that is the SO splitting in the carbon isotopes, we investigate the evolution of one-proton separation energies for carbon and oxygen isotopes. In Fig. 5, the horizontal bars represent the experimental one-proton addition ($\epsilon_{+p}$) and removal ($\epsilon_{-p}$) energies for (a) carbon and (b) oxygen isotopes deduced from one-proton separation energies, that are binding energies of boron to fluorine isotopes, and the excitation energies of the lowest $3/2^-$ states in the odd–even nitrogen isotopes. The dotted bars indicate the adopted values for the observed excited states in $^{19,21}$N, which have been tentatively assigned as $3/2^{-57}$. Other experimental data are taken from refs. [31,58,59]. For comparison, we show the one-proton addition and removal energies (blue symbols) calculated using the shell model with the YSOX interaction[60], which was constructed from a monopole-based universal interaction ($V_{MU}$). Because the phenomenological effective two-body interactions were determined by fitting experimental data, they are expected to partially include the three-nucleon effect and thus can reproduce relatively well the ground-state energies, drip lines, energy levels, as well as the electric and spin properties of carbon and oxygen isotopes. As

shown in Fig. 5, the shell model calculations reproduce the $\epsilon_{\pm p}$'s for carbon and oxygen isotopes very well.

As mentioned earlier, in the absence of many-body correlations, $\epsilon_{\pm p}$ resemble the proton single-particle energies, and the gap between them can be taken as the (sub)shell gap. In the following, we consider $^{14,15}$C and the closed-shell $^{14,16,22}$O isotopes in more detail. We computed their ground-state binding energies and those of their neighbouring isotones $^{13,14}$B, $^{13,15,16,21}$N and $^{15,17,23}$F. We applied the Λ-CCSD(T) and the particle-attached/removed EOM-CC methods to compute the binding energies for the closed-(sub)shell and open-shell nuclei, respectively. The ground-state binding energies of $^{14}$B ($2^-$) and $^{16}$N ($2^-$) were computed using the EOM-CC method with reference to $^{14}$C and $^{16}$O employing the charge-exchange EOM-CC technique[61]. Results of the CC calculations on $^{14,15}$C and $^{14,16,22}$O with and without 3NFs are shown by the red-solid and red-dashed lines, respectively. Here, we have opted for EM(1.8/2.0 fm$^{-1}$), which yield the smallest chi-square value for the calculated and experimental binding energies considered, as the NN+3NF interactions. For the NN-only interaction, we show the calculations with EM(2.8 fm$^{-1}$). The calculated $\epsilon_{-p}(3/2^-)$ for $^{22}$O with EM(2.8 fm$^{-1}$) (and other NN-only interactions) has an unrealistic positive value, and is thus omitted. We found the norms of the wave functions for the one-particle (1p) $1/2^-$ and one-hole (1h) $3/2^-$ states of $^{14}$C, and the two corresponding 1p and 1h states of $^{15}$C ($2^-$ states in $^{14}$B and $^{16}$N) to be almost 90%. The calculations suggest that these states can be accurately interpreted by having dominant single-particle structure, and that the gaps between these 1p–1h states resemble the proton-subshell gaps. It is obvious from the figure that the calculations with the NN + 3NF interactions reproduce the experimental $\epsilon_{\pm p}$ for

$^{14,15}$C and $^{14,16,22}$O very well. Overall, the calculations without 3NFs predict overbound proton states, and in the case of $^{14,15}$C, much reduced subshell gaps. These results suggest that $^{14}$C is a doubly-magic nucleus, and $^{15}$C a proton-closed shell nucleus.

Our results show that the phenomenon of large spin–orbit splitting is indeed universal in atomic nuclei, and the magic number 6 is as prominent as other classical SO-originated magic numbers such as 28. Although we have shown only results for $^{14,15}$C, we expect further systematic and detailed theoretical analyses on other carbon isotopes, in particular ab initio calculations using realistic and/or chiral interactions, to provide quantitative insights on the neutron-number dependence of the SO splitting and its origin. It will be interesting to understand also the origins of the diverse structures in $^{12}$C.

Finally, we would like to point out that an inert $^{14}$C core, built on the $N = 8$ closed shell, has been postulated to explain several experimental data for $^{15,16}$C. For instance, a $^{14}$C + $n$ model was successfully applied[62] to explain the consistency between the measured g-factor and the single-particle-model prediction (the Schmidt value) of the excited $5/2^+$ state in $^{15}$C. Wiedeking et al.[25], on the other hand, have explained the small $B(E2)$ value in $^{16}$C assuming a $^{14}$C + $n$ + $n$ model in the shell-model calculation. In terms of spectroscopy studies using transfer reactions, the results from the $^{14}$C($d$, $p$)$^{15}$C[63] and $^{15}$C($d$, $p$)$^{16}$C[64] measurements are also consistent with the picture of a stable $^{14}$C core. On the proton side, a possible consolidation of the $1p_{3/2}$ proton-subshell closure when moving from $^{12}$C to $^{14}$C was reported decades ago from the measurements of the proton pick-up ($d$,$^3$He) reaction on $^{12,13,14}$C targets[65], consistent with shell model predictions. An attempt to study the ground-state configurations with protons outside the $1p_{3/2}$ orbital in $^{14,15}$C has also been reported[66] very recently. To further investigate the proton-subshell closure in the neutron-rich carbon isotopes, more experiments using one-proton transfer and/or knockout reactions induced by radioactive boron, carbon and nitrogen beams at facilities such as ATLAS, FAIR, FRIB, RCNP, RIBF and SPIRAL2 are anticipated.

## Methods

**Experiment and data analysis.** Secondary $^{12-18}$C beams were produced, in separate runs, by projectile fragmentation of $^{22}$Ne$^{10+}$ ions at 80 MeV per nucleon incident on a $^9$Be (production) target with thickness ranging from 1.0 to 5.0 mm. The carbon beam of interest was selected by setting the appropriate particle magnetic rigidities using the RCNP EN fragment separator. The carbon beam thus produced was transported to the experimental area, and directed onto a 450-mg cm$^{-2}$-thick natural carbon (reaction) target. The incident beam was identified by the measurements of energy loss in a 320-mm-thick silicon detector, and the time of flight (TOF) between the production and reaction targets. The TOF was determined from the timing information obtained with a 100-μm-thick plastic scintillation detector placed before the reaction target and the radio-frequency signal from the accelerator. Particles exiting the reaction target were detected by a multisampling ionisation chamber (MUSIC), consisting of eight anodes and nine cathodes, before being stopped in a 7-cm-thick NaI(Tl) scintillation detector. The outgoing particles were identified using the energy-loss and total-energy information obtained with the MUSIC and NaI(Tl) detectors. Data acquisition was performed using the software package babirlDAQ[67]. The charge-changing cross sections were measured using the transmission method taking into account the geometrical acceptance of the MUSIC and NaI(Tl) detectors. In the present transmission method, the numbers of incident carbon beam and outgoing carbon particles, including lighter carbon isotopes, were identified and counted.

**Proton radii and Glauber-model analysis.** The point-proton root-mean-square radius is defined as follows:

$$R_{\mathrm{P}} \equiv \left\langle r_{\mathrm{P}}^2 \right\rangle^{1/2} = \left( \int \rho_{\mathrm{P}}(\mathbf{r}) r^2 \mathrm{d}\mathbf{r} \right)^{1/2}, \qquad (2)$$

where $\rho_{\mathrm{P}}(\mathbf{r})$ is the proton density distribution, $\mathbf{r}$ is the radial vector, and $r$ is the radius. To extract the proton radii from the measured charge-changing cross sections, we performed reaction calculations using the recently formulated Glauber model[37] within the optical-limit approximation. We assumed that charge-changing cross section depends only on the proton density distribution in the carbon

projectile. By adopting a simple one-parameter HO or a two-parameter Woods–Saxon (WS) density distribution for the protons, we determined the parameter(s) so as to reproduce the experimental data. $R_{\mathrm{p}}$ is then calculated by substituting the obtained proton density distribution into Eq. (2). The difference (about 0.5%) between the $R_{\mathrm{p}}$ values determined with different functional forms was taken as the systematic uncertainty. The HO-type and WS-type density distributions are given by:

$$\rho_{\mathrm{P}}^{\mathrm{HO}}(Z, R_{\mathrm{HO}}, r) = \rho_0^{\mathrm{HO}} \exp\left[-\left(\frac{r}{R_{\mathrm{HO}}}\right)^2\right]\left[1 + \frac{Z-2}{3}\left(\frac{r}{R_{\mathrm{HO}}}\right)^2\right],$$

$$\rho_{\mathrm{P}}^{\mathrm{WS}}(R_{\mathrm{WS}}, a, r) = \rho_0^{\mathrm{WS}}\left[1 + \exp\frac{(r-R_{\mathrm{WS}})}{a}\right]^{-1}$$

where $\rho_0^{\mathrm{HO}}$ and $\rho_0^{\mathrm{WS}}$ are the central densities, which are uniquely determined by the conservation of proton number ($Z$). $R_{\mathrm{HO}}$ is the HO width parameter, while the parameters $R_{\mathrm{WS}}$ and $a$ are the half-density radius and diffuseness, respectively.

**Data availability**. The data that support the findings of this study are available from the corresponding author upon reasonable request.

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

# ARTICLE

26. Ong, H. J. et al. Lifetime measurements of first excited states in $^{16,18}$C. *Phys. Rev. C* **78**, 014308 (2008).
27. Garcia Ruiz, R. F. et al. Unexpectedly large charge radii of neutron-rich calcium isotopes. *Nat. Phys.* **12**, 594–598 (2016).
28. Wienholtz, F. et al. Masses of exotic calcium isotopes pin down nuclear forces. *Nature* **498**, 346–349 (2013).
29. Angeli, I. & Marinova, K. P. Correlations of nuclear charge radii with other nuclear observables. *J. Phys. G* **42**, 055108 (2015).
30. Pritychenko, B., Birch, M., Singh, B. & Horoi, M. Tables of E2 transition probabilities from the first $2^+$ states in even–even nuclei. *At. Data Nucl. Data Tables* **107**, 1–055139 (2016).
31. Wang, M. et al. The AME2016 atomic mass evaluation. *Chin. Phys. C* **41**, 030003 (2017).
32. Terashima, S. et al. Proton radius of $^{14}$Be from measurement of charge-changing cross sections. *Prog. Theor. Exp. Phys.* 101D02 (2014).
33. Estrade, A. et al. Proton radii of $^{12-17}$B define a thick neutron surface in $^{17}$B. *Phys. Rev. Lett.* **113**, 132501 (2014).
34. Kanungo, R. et al. Proton distribution radii of $^{12-19}$C illuminate features of neutron halos. *Phys. Rev. Lett.* **117**, 102501 (2016).
35. Suzuki, Y. et al. Parameter-free calculation of charge-changing cross sections at high energy. *Phys. Rev. C* **94**, 011602(R) (2016).
36. Shimoda, T. et al. Design study of the secondary-beam line at RCNP. *Nucl. Instrum. Methods B* **70**, 320–330 (1992).
37. Tran, D. T. et al. Charge-changing cross-section measurements of $^{12-16}$C at around 45A MeV and development of a Glauber model for incident energies 10A–2100A MeV. *Phys. Rev. C* **94**, 064604 (2016).
38. Geng, L. S., Toki, H. & Meng, J. Masses, deformations and charge radii—nuclear ground-state properties in the relativistic mean field model. *Prog. Theor. Phys.* **113**, 785–800 (2005).
39. Ekstrom, A. et al. Accurate nuclear radii and binding energies from chiral interaction. *Phys. Rev. C.* **91**, 051301(R) (2015).
40. Ekstrom, A. et al. Optimized chiral nucleon–nucleon interaction at next-to-next-to-leading order. *Phys. Rev. Lett.* **110**, 192502 (2013).
41. Collard, H. R., Elton, L. R. B. & Hofstadter, R. *Nuclear radii* 2 (Springer, Berlin, 1967).
42. Bohr, A. & Mottelson, B. R. *Nuclear Structure: Single Particle Motion* 1 (World Scientific, Singapore, 2008).
43. Wigner, E. P. & Feenberg, E. Symmetry properties of nuclear levels. *Rep. Prog. Phys.* **8**, 274–317 (1941).
44. Chasman, R. R. n-p pairing, Wigner energy, and shell gaps. *Phys. Rev. Lett.* **99**, 082501 (2007).
45. Cohen, S. & Kurath, D. Spectroscopic factors for the 1p shell. *Nucl. Phys. A.* **101**, 1–16 (1967).
46. Cohen, S. & Kurath, D. Effective interactions for the 1p shell. *Nucl. Phys.* **73**, 1–24 (1965).
47. Taube, A. G. & Barlett, R. J. Improving upon CCSD(T): ΛCCSD(T). I. Potential energy surfaces. *J. Chem. Phys.* **128**, 044110 (2008).
48. Gour, J. R. et al. Coupled-cluster calculations for valence systems around $^{16}$O. *Phys. Rev. C* **74**, 024310 (2006).
49. Morris, T. D. et al. Structure of the lightest tin isotopes. Preprint at https://arxiv.org/abs/1709.02786 (2017).
50. Hebeler, K. et al. Improved nuclear matter calculations from chiral low-momentum interactions. *Phys. Rev. C* **83**, 031301(R) (2011).
51. Entem, D. R. & Machleidt, R. Accurate charge-dependent nucleon–nucleon potential at fourth order of chiral perturbation theory. *Phys. Rev. C* **68**, 041001 (R) (2003).
52. Bogner, S. K., Furnstahl, R. J. & Perry, R. J. Similarity renormalization group for nucleon–nucleon interactions. *Phys. Rev. C* **75**, 061001(R) (2007).
53. Hagen, G. et al. Neutron and weak-charge distributions of the $^{48}$Ca nucleus. *Nat. Phys.* **12**, 186–190 (2016).
54. Otsuka, T. et al. Three-body forces and the limit of oxygen isotopes. *Phys. Rev. Lett.* **105**, 032501 (2010).
55. Hagen, G. et al. Continuum effects and three-nucleon forces in neutron-rich oxygen isotopes. *Phys. Rev. Lett.* **108**, 242501 (2012).
56. Cipollone, A., Barbieri, C. & Navratil, P. Isotopic chains around oxygen from evolved chiral two- and three-nucleon interactions. *Phys. Rev. Lett.* **111**, 062501 (2013).
57. Sohler, D. et al. In-beam γ-ray spectroscopy of the neutron-rich nitrogen isotopes $^{19-22}$N. *Phys. Rev. C* **77**, 044303 (2008).
58. Ajzenberg-Selove, F. Energy levels of light nuclei A = 13–15. *Nucl. Phys. A* **523**, 1–196 (1991).
59. Tilley, D. R., Weller, H. R. & Cheves, C. M. Energy levels of light nuclei A = 16–17. *Nucl. Phys. A* **565**, 1–184 (1993).
60. Yuan, C., Suzuki, T., Otsuka, T., Xu, F.-R. & Tsunoda, N. Shell-model study of boron, carbon, nitrogen and oxygen isotopes with a monopole-based universal interaction. *Phys. Rev. C* **85**, 064324 (2012).
61. Ekstrom, A. et al. Effects of three-nucleon forces and two-body currents on Gamow-Teller strengths. *Phys. Rev. Lett.* **113**, 262504 (2014).
62. Hass, M., King, H. T., Ventura, E. & Murnick, D. E. Measurement of the magnetic moment of the first excited state of $^{15}$C. *Phys. Lett. B59*, 32–34 (1975).
63. Goss, J. D. et al. Angular distribution measurements for $^{14}$C(d, p)$^{15}$C and the level structure of $^{15}$C. *Phys. Rev. C* **12**, 1730–1738 (1975).
64. Wuosmaa, A. H. et al. $^{15}$C(d, p)$^{16}$C reaction and exotic behavior in $^{16}$C. *Phys. Rev. Lett.* **105**, 132501 (2010).
65. Mairle, G. & Wagner, G. J. The decrease of ground-state correlations from $^{12}$C to $^{14}$C. *Nucl. Phys. A* **253**, 253–262 (1975).
66. Bedoor, S. et al. Structure of $^{14}$C and $^{14}$B from the $^{14,15}$C(d, $^3$He)$^{13,14}$B reactions. *Phys. Rev. C* **93**, 044323 (2016).
67. Baba, H. et al. New data acquisition system for the RIKEN radioactive isotope beam factory. *Nucl. Instrum. Methods A* **616**, 65–68 (2010).

## Acknowledgements

We thank T. Shima, H. Toki, K. Ogata and H. Horiuchi for discussion, K. Hebeler for providing matrix elements in Jacobi coordinates for 3NFs at NNLO, and C. Yuan for comments. H.J.O. and I.T. thank A. Tohsaki and his spouse, D.T.T. and T.T.N. acknowledge RCNP Visiting Young Scientist Support Program, D.T.T. and T.H.H. thank Nishimura International Scholarship Foundation and Matsuda Yosahichi Memorial Foreign Student Scholarship, respectively, for support. This work was supported in part by Hirose International Scholarship Foundation, the JSPS-VAST Bilateral Joint Research Project, Grand-in-Aid for Scientific Research Nos. 20244030, 20740163 and 23224008 from Japan Monbukagakusho, the Office of Nuclear Physics, U.S. Department of Energy, under grants DE-FG02-96ER40963, DE-SC0008499 (NUCLEI SciDAC collaboration), the Field Work Proposal ERKBP57 at Oak Ridge National Laboratory (ORNL), and the Vietnam government under the Program of Development in Physics by 2020. Computer time was provided by the Innovative and Novel Computational Impact on Theory and Experiment (INCITE) program. This research used resources of RCNP Accelerator Facility and the Oak Ridge Leadership Computing Facility located at ORNL, which is supported by the Office of Science of the Department of Energy under Contract No. DE-AC05-00OR22725.

## Author contributions

H.J.O. initiated the project, performed systematic analysis, prepared the figures and wrote the manuscript. D.T.T. performed data analysis and Glauber-model calculations. G.H., T. D.M. and G.R.J. performed the CC calculations. T.S. and T.O. performed the shell model calculations. Y.K.-E. and L.S.G. performed the AMD and RMF calculations, respectively. D.T.T., H.J.O., N.A., S.T., I.T., T.T.N., Y.A., P.Y.C., M.F., H.G., M.N.H., T.H., T.H.H., E.I., A.I., R.K., T.K., L.H.K., W.P.L., K.M., M.M., S.M., D.N., N.D.N., D.N., A.O., P.P.R., H.S., C.S., J.T., M.T., R.W. and T.Y. performed the experiments. All authors discussed and commented on the manuscript.

## Additional information

**Competing interests:** The authors declare no competing interests.

