## [Peer Review File · Nature Communications]

REVIEWERS' COMMENTS:

Reviewer #2 (Remarks to the Author):

The authors have updated the manuscript based on the reports from the three referees, but this has resulted in only a slight change in presentation. What remains is a interesting enough analysis of a specialised area that has already received considerable attention. It remains as the original submission, something at the level of a regular contribution to the canon for specialist nuclear physicists. I find it hard to recommend as a showpiece of our field to the rest of the physics community.

Reviewer #3 (Remarks to the Author):

The authors have reasonably answered the comments on the original manuscript which was submitted to Nature. In the revised version which is transferred to Nature communications, the authors have improved their expressions with emphasizing the experimental evidences for the $Z=6$ magic number and as well correcting the definition of shell gap [i.e., Eq.(1)]. The present manuscript seems to be suitable for publication in Nature communications.

Reply to Reviewer #2

Authors: D.T.Tran, H.J. Ong (corresponding author) et al.

Title: Evidence for prevalent $Z=6$ magic number in neutron-rich carbon isotopes.

- **Comment** *The authors have updated the manuscript based on the reports from the three referees, but this has resulted in only a slight change in presentation. What remains is a interesting enough analysis of a specialised area that has already received considerable attention. It remains as the original submission, something at the level of a regular contribution to the canon for specialist nuclear physicists. I find it hard to recommend as a showpiece of our field to the rest of the physics community.*

Reply:

We thank the reviewer for the time and valuable comments, and for considering the results of our analysis interesting. While we regret our inability to convince the reviewer, we remain optimistic that the finding of this prevalent and smallest spin-orbit-originated shell closure will provide important benchmarks for nuclear-structure theories. The present result is important to stimulate more concerted effort towards understanding the microscopic origins of the large spin-orbit splittings, which are shown here, at last, to be universal in atomic nuclei.

Reply to Reviewer #3

Authors: D.T.Tran, H.J. Ong (corresponding author) et al.

Title: Evidence for prevalent $Z=6$ magic number in neutron-rich carbon isotopes.

- **Comment** *The authors have reasonably answered the comments on the original manuscript which was submitted to Nature. In the revised version which is transferred to Nature communications, the authors have improved their expressions with emphasizing the experimental evidences for the $Z=6$ magic number and as well correcting the definition of shell gap [i.e., Eq.(1)]. The present manuscript seems to be suitable for publication in Nature communications.*

Reply:

We thank the reviewer for the time, and for recommending our manuscript for publication.

The valuable comments from the reviewer have certainly helped to improve our manuscript.